# The Role of Maternal Grandmothers’ Childcare Provision for Korean Working Adult Daughters

**DOI:** 10.3390/ijerph192114226

**Published:** 2022-10-31

**Authors:** Sesong Jeon, Katie Walker

**Affiliations:** 1Major in Child & Family Studies, School of Child Studies, College of Human Ecology, Kyungpook National University, Daegu 41566, Korea; 2Children’s Studies, Child Life and Health Program, Eastern Washington University, Cheney, WA 99004, USA

**Keywords:** childcare, working adult daughter, intergenerational solidarity model, ambivalence

## Abstract

Despite the Korean government’s investment in childcare facilities for dual-earner households, maternal grandmothers are increasingly taking on the responsibility of caring for their grandchildren. This trend is examined in the current research. While many studies have been conducted on grandparents’ experiences providing childcare for their grandchildren, significantly less research has been conducted on adult daughters’ experiences with their mothers’ childcare provision. This study utilized the concepts of intergenerational solidarity and a life-course approach to understand the experiences of 24 working adult daughters in Korea (ages 30–43) whose mothers provide childcare. Three major themes were identified following a grounded theory approach: gratitude vs. guilt, dependence vs. independence, and closeness vs. disagreement. The results indicated that adult daughters were found to have ambivalence toward their mothers, reflecting the lack of alternative options for childcare. The results from this study suggest that not only improving the quality of public childcare services, but also diversifying services to reflect the needs of dual-income families.

## 1. Introduction

Childcare provided by grandparents is a global phenomenon, and it contributes to a growing body of literature on intergenerational relationships [1]. Recently, the new words “halma” (grandmother + mommy) and “halpa” (grandfather + daddy) have emerged in Korea, which refers to grandparents who provide childcare for their working adult children. In Korea, grandmothers provide the majority of grandchild care, and maternal grandmothers account for more than half of grandmothers’ childcare (56.8%) [2]. Research has found that childcare can positively or negatively impact the intergenerational relationship between parents and their adult children [3,4].

To date, socio-psychological research on grandparent childcare has focused on: (1) psychological well-being of grandparents and (2) factors associated with psychological well-being of grandparents. The first of these aspects heavily focuses on the psychological experience of grandparents as a primary caregiver [5]. While researchers have focused extensively on the mental health of grandparents in the role of primary caregiver, they have paid less attention to the perspectives of adult children. Furthermore, many studies shed light on the importance of social support on the grandparents’ psychological well-being [6]. Less attention has been given to what factors between adult children and their parents affect both parties’ psychological well-being.

Similar to global studies on the provision of childcare, Korean researchers also have mainly conducted studies on primary grandchild care [7]. Furthermore, qualitative research has been relatively under-utilized, with most of the research focusing on grandparents’ childcare experience [8]. The unusual dynamics between working-women and their mothers who provided childcare have not been fully explained in the few studies that have studied adult children’s perspective when grandparents provided childcare in Korea [9]. (i.e., working adult daughter and daughter-in-law; working adult daughter and non-working adult daughter). Moreover, the studies that have considered Korean adult daughters’ perspectives took place before the Universal Childcare Subsidy (In 2013, because of the growth of dual-income families, the Korean government implemented a free childcare policy as part of a policy effort to prevent the absence of childcare. The Universal Childcare Subsidy allows all households to have full-day center-based care or home childcare allowance for children 0–5 years old regardless of income status) was enacted in 2013. Both studies [9,10] collected data before a Universal Childcare Subsidy on childcare in Korea was implemented (2007–2008).

In order to fill the gap, this study not only utilizes a qualitative approach to deepen the fields’ understanding of working adult daughters’ childcare experience but also investigates whether the current policy on childcare impacts the relationship quality between grandparents and their adult children; therefore, the data for the current research were collected in 2016–2017.

## 2. Theoretical Framework

Guided by the life course perspective and the intergenerational solidarity model, the current study examines how working adult daughters’ experience their current relationship with their mothers who provide supplementary grandchild care. First, the concept of *linked lives* within the life course perspective will be applied to understand the mother-daughter relationship. Linked lives is defined as a unit of family members composed of a set of mutually interlocking lives whose members share and develop life opportunities in their families over time [11]. Korean working adult daughters tend to depend on their parents despite parents-in-law living closer to them [12]. Korean aging mothers also view providing childcare as an investment in their daughters [13]. Although both mothers and their working adult daughters might have different motivations for childcare, they are linked through the experience of childcare and, which may lead to spending more time together. This, in turn, may impact the adult daughters’ perceptions of the relationship quality with their mothers who provide childcare because they are influencing each other.

Second, relationship quality between mothers and their daughters in adulthood will be considered in the Korean context. The social structural shift in workforce participation rate of the daughters’ generation is nearly double that of the mothers’ generation. This shows the drastic change in the workforce at the macro level in a Korean family. In Korea, the rate of female labor force participation increased from 26.8% in 1960 to 52.7% in 2017 [14]. This may contribute to a different perspective of the workforce and the meaning of childcare between mothers and their adult daughters.

Furthermore, mothers and their daughters are living in a society that has become more contemporary. The mothers have lived through both traditional values and the new Western values, whereas the daughters have mostly experienced Western values. This suggests that mothers and daughters have experienced different roles in the family. Mothers have lived through both the traditional values and the new Western values after the Korean War [15]. Most aging Korean mothers believe that being a mother means they will become one with their children [16], and their goal in life is closely connected with their children. On the other hand, younger Korean mothers have known only the post-war values influenced by urbanization and industrialization [17]. Therefore, Korean daughters who were raised using a westernized education system have learned the importance of differentiation and independence. Korean society is highly influenced by the individualistic culture; however, the high percentage of childcare by grandparents indicates that Korea is still a collectivist culture [18]. Thus, the shift in cultural values in Korea may affect the dynamics of the relationship between working adult daughters and mothers who provide childcare.

The intergenerational solidarity model introduced by Bengtson [19] has served as a key framework to explore parent–child relationships worldwide [20]. Intergenerational relations can be assessed through six dimensions: (1) affectual solidarity (e.g., positive or negative emotions), (2) associational solidarity (e.g., frequency and pattern of interactions), (3) consensual solidarity (e.g., agreement on values), (4) functional solidarity (e.g., exchange of support/resources), (5) normative solidarity (e.g., feelings of family obligation), and (6) structural solidarity (e.g., geographic factors). In regard to parent–child relations, supplemental grandparent childcare has been a relevant topic in the intergenerational solidarity model, but less is known about perceptions of working adult children when compared to grandparents’ perspectives. The majority of studies utilizing the solidarity model have focused on how solidarity between older parents and adult children has impacted the parents’ psychological well-being [21].

Although studies have demonstrated the positive role of solidarity between parents and their adult children in the context of providing childcare, childcare produced conflict [22]. Furthermore, when exploring grandparents’ childcare provision, scholarly research has yielded mixed results, with grandparents experiencing both positive and negative emotions [8,23]. Few studies have explored adult children’s feelings of ambivalence when grandparents provided childcare in Korea. Lee [10] explored the relationship quality between grandparents and adult children. The findings indicated that grandparents felt ambivalence, while on the other hand, adult children felt satisfied [10]. Kim et al. [9] also found that adult children reported feeling ambivalent towards grandparents, with working-women reporting higher ambivalence than non-working women. However, these two studies did not fully incorporate Korea’s changed childcare policies, which may improve grandparents’ relationships with adult children. With a limited number of studies exploring how childcare is involved in the complicated relationship contexts between parents and their adult children, the current research uses the intergenerational solidarity model to further explore the perceptions of working adult daughters in Korea.

To sum up, the current study utilizes the life course perspective and intergenerational solidarity model in order to investigate how working adult daughters perceive their relationship with their mothers who provide childcare in Korea by using a standard qualitative methodological approach. Investigating the perception of adult daughters who are balancing the relationship between western and traditional Korean values will advance the field of intergenerational relationship research. The research question is as follows: How do Korean working adult daughters describe their current relationship with their mothers who provide supplementary grandchild care?

## 3. Method

### 3.1. Participants

Twenty-four working adult daughters (ages 30–43) living in Korea (1) who have at least one young child (0–5 years old) under the Universal Childcare Subsidy and (2) who rely on their mothers for child care were the targeted study population for in-depth interviews.

### 3.2. Procedure

With approval from Institutional Review Board at Iowa State University in 2015, participants were recruited in Korea. To find participants who fit the selection criteria, multiple recruitment strategies were used such as flyers, informal talks, and website for advertising. A snowball sample technique was mainly utilized. The first interviewee was introduced by a Korean professional collegue from Kyungpook National University, Daegu, in South Korea. Some of the interviewees introduced participants who have the same occupation with them; however, working women with various types of occupation were selected to reduce sampling bias.

For the interview, they were contacted via e-mail, text message, or phone and asked to participate in an interview about their relationship with their mothers. The interviews were voluntary, and no compensation for their participation in this study was offered. After recruitment, semistructured interviews were conducted between 2016 and 2017 in Korea (summer 2016, winter 2016, and summer 2017). The interviews lasted 1–2 h (average 90 min) and were audio recorded. Participants selected the location and time of the interview. They decided upon a private room at the library, coffee shop, or their homes. Before the interview, the study procedures were provided with participants and then informed consent form was obtained from the participants.

### 3.3. Analysis

Semi-structured interviews were conducted in this current study. The open-ended questions focused on the interpersonal, familial, and contexual process of Korean working adult daughters’ perception of the relationship with their mothers who provide childcare. The questions respondents were asked are as follows: warm-up and informational questions (i.e., age, income, and date of birth, etc.); relationship with mothers (i.e., past & current relationship with mother); childcare supports from other family members (i.e., husband, siblings, and family-in-law); childcare in Korea (i.e., maternity leave, compensation of childcare).

To analyze the data, Max-QDA 12.0 qualitave software was utilized. Before using the Max-QDA program, the data were reorganized: (1) the voice recorded data were transcribed into Korean using MS Word; and (2) the document was translated into English again. To check for reliability, two female research assistants who are native Korean speakers at Iowa State University translated 24 cases into English. They also back-translated some cases. If we had different translations, we discussed them until achieving consensus.

After these initial processes, the translated interview data were imported into the Max-QDA program. Next, we followed a standard qualitative methodological approach involving a three-step coding process [24]. Before the coding process, we read each transcript several times to ensure complete understanding. First, “line-by-line” coding was performed to encapsulate scattered data. Some content was highlighted with different colors and memos were created in MAX-QDA. Second, “focused” coding was utilized to create related categories by linking initial line-by-line codes. Finally, themes from the related categories [24] were created. We used the “immersion /crystallization” method, which is a consensus approach for discussing discrepancies in codes until consensus is reached.

## 4. Findings

The average age of the working adult daughters was 35.71 years (SD = 2.91) and their mothers’ mean age was 62.42 years (SD = 4.75). Total monthly income in the household ranged from KRW (KRW refers to the won, the Korean currency. Using the exchange rate in effect at the time of data collection, KRW 1145 was equivalent to roughly $1 USD) 3,500,000 ($3056) to KRW 11,750,000 ($10,262). Their jobs were varied, for example: nurse, social worker, teacher, businessman, police officer, trade officer, and child pychologist. Table 1 provides further demographic information of respondents. As a way to guarantee the protection of the participants, we used pseudonyms.

The working adult daughters described their current relationship with their mothers who provided childcare with specific personal events. The current study found the six aspects of the intergenerational solidarity model. The results indicated that several dimensions were interconnected. The three emergent themes are as follows; (1) gratitude vs. guilt, (2) dependence vs. independence, and (3) closeness vs. disagreement. The three emerged themes also supported the linked lives perspective. The adult daughters felt ambivalence toward their mothers who provided support and this indicates they are influencing each other.

### 4.1. Gratitude vs. Guilt

Adult daughters’ relationship satisfaction with their mothers, mainly expressed as gratitude to their mothers, was due to help with childcare. Nineteen adult daughters out of 24 were satisfied with their current relationship with their mothers and felt thankfulness. For example, Joo-young, who could resume her work after her delivery, appreciated her mother’s childcare support:


*I appreciate my mother’s sacrifice. If there was no sacrifice from my mom, I could not work. It means that I might not have any financial ability. Therefore, her sacrifice is also helpful for my financial status. Of course, the government supports my baby for $150 per month. To my knowledge, it will continue only 2 years. I don’t think it’s helpful. Actually, it is my social right due to paying tax. Rather than the help, my mother supports me a lot. It is a great thing that there was no cut off time in my career and it’s all due to my mom.*


Similarly, adult daughters felt gratitude to their mothers due to increased understanding of the mothers after childbirth (*N* = 15). Furthermore, the adult daughters felt gratitude to their mothers because they perceived positive relationship quality between their mothers and other family members (*N* = 12). They described that the relationship between not only their husbands and their mothers, but also their children and their mothers, were good. Their perceived positive relationships were connected with adult daughters’ thankfulness toward their mothers. For example, Seo-jin provided the following description of the relationship between her husband and her mother: “My mother and husband get along very well. Even better than me. If my mother has a conflict with me, she always talks about it with my husband […] It’s really good to see them get along well. I appreciate my mom for this”. Yoon-a also positively described the relationship between her mother and her daughter:


*When my mother and I are together, my daughter goes to my mother. I do not feel bad at all, but thankful. If the person who raises my child does not get along well, I will feel worried. However, since my mom and my daughter get along very well, I feel thankful for my mom.*


In contrast, adult daughters who reported negative relationships with their mothers described a difficult time to communicate (*N* = 3). Often these adult daughters described issues in their relationships with their mothers since childhood. These daughters described their inevitable relationship with their mothers due to the lack of alternatives of childcare. For example, Dan-i, who asks her mother to provide childcare as needed instead of on a regular schedule, was not satisfied with the relationship with her mother:


*I don’t have good memories with my mother when I was young. No happy, warm and exciting memory with my mom.[…] From here, I think she reminds me of…when I was young. It makes me upset that I got shortage of parenting.[…] I do not leave my child to her always, but she cares for my child only when I have conference for which I need to travel or if there is an emergency at work. I seldom ask her to take care of my children.*


Despite these negative mother-adult daughter relationships, some adult daughters indicated that they hoped the support from their mothers would continue until their children grew up (*N* = 3). Meanwhile, the adult daughters’ gratitude toward their mothers was associated with feelings of guilt (*N* = 16). There are three main sources of guilt. First, the lack of personal time for their mothers due to providing childcare made the adult daughters feel guilty (*N* = 7). For example, Young-mi, who has a 2-year old son with special needs on linguistic development, appreciated her mother who not only takes the son to the speech therapy center but also feeds him, even on weekends. However, she felt sorry for her mother because of the mother’s lack of leisure time. Similarly, Na-ra, whose mother takes care of her daughter for 11 h a day, shared similar feelings:


*I feel great gratitude to my mom, but probably she might not be satisfied with our relationship because she cannot have her free time much. I feel sorry for my mom. She had a hard time raising us (3 children), and now she is doing parenting again because of my child. Although I got allowance from government, it’s not enough. I hope to implement a policy for supporting grandparents who are taking care of grandkids like my mom. She is getting old, and she should have her own time, but she cannot. So, I feel so sorry and sad.*


Second, adult daughters felt guilty for their mothers’ health deterioration due to childcare (*N* = 6). Se-ri, whose mother takes care of her two little daughters, felt gratitude for her mother who provides frequent additional childcare during her busy time. At the same time, she expressed feeling guilty:


*[…] Thankfully, I could extend my work due to my mom’s support. Unfortunately, I have to do extra work frequently. Even I got universal childcare subsidy, I don’t think it works well. I prefer my mom to take care of my baby because my mom can help me whenever I have extra work. I believe that the government should make better work environment first rather than giving us small amount of money. […] My mom is about 70 years old but still carries babies on her back and it harms her lower back which needs care. So, I feel sorry for her condition in her lower back which is damaged by caring for my babies. I think it’s my fault.*


Third, adult daughters felt guilty because they did not express gratitude towards their mothers (*N* = 3). Seo-jin, who had already received childcare for her first daughter from her mother, felt continuous gratitude towards her mother who is taking care of her second daughter. However, she was sorry for not expressing thankfulness to her mother when she reflected on her behavior so far. Min-a, who is living with her mother, also felt guilty for failing to express her appreciation despite her gratefulness for her mothers’ help:


*[…] My mother has taken care of my two children. I couldn’t get free childcare service for my first child but now my second child got the benefits. Well… That’s better to have it rather than nothing. I much more appreciate my mother. Best of all, I appreciate my mother who is waiting for me after I get tired of coming back from work. She prepared my meals. […] I wish I could express my gratitude to my mom more often. Even though I am consciously aware of it, I am sorry for my mom.*


She shed tears during the interview and continued to express her guilt toward her mother.

In sum, adult daughters generally appreciated their mothers and reported maintaining a satisfactory relationship with their mothers. However, at the same time the adult daughters reported feeling guilt toward their mothers. The mixed relational satisfaction with their mother fits well with the concept of affectional solidarity; however, gratitude and guilt did not perfectly match any dimensions of solidarity. The emerged theme might reflect a different type of solidarity. In addition, the adult daughters mentioned childcare policy when they described the current relationship with their mothers. Few adult daughters were satisfied with current childcare policy.

### 4.2. Dependence vs. Independence

Participants expressed feelings of dependence on their mothers, however, they also desired independence. Adult daughters expressed dependence on their mothers when describing the importance of their mothers in their lives. The adult daughters were particularly dependent on childcare, and many daughters said that they would not be able to care for their children without their mothers’ help (*N* = 15). There were even adult daughters who expressed that their mothers were more important persons than their husbands (*N* = 9). Adult daughters often described that the dependence on their mothers is connected with their full-time careers (*N* = 10). Se-ri, who is a director of the kindergarten, stated her ability to work full-time came from relying on her mother’s support:


*If my mom were not with me, my life wouldn’t exist either. I also could not study. I currently completed a doctoral course and finished my master’s degree after I got married. If my mom doesn’t support me in child-rearing, I won’t be able to complete all studies…. To express my gratitude toward her, I give allowance to her every month… but it’s not a big amount though.*


Furthermore, some adult daughters showed high emotional dependence on their mothers (*N* = 9). For example, Seo-jin, who has experienced high dependence on her mother since before her marriage, described life-long emotional reliance on her mother:


*I grew up under my mother’s rule, and I never got out of that frame. […] In my college life, I tried to escape from my mom’s guideline, but it did not work that long. […] Until now, if I have to decide on something, I asked her to go outside serving dinner to get advice. Sometimes we spent a whole day together… after dinner we had a tea time or watched new movie. She wanted to pay for them, however, I paid all expenses…because I got a lot of advice from her. I rely on my mom regarding a lot in making a decision. She makes me happy to solve some issue. I just have lived in her guidelines, and I do not have any complaints.*


Another adult daughter, Dan-i, stated her financial dependence on her mother. Although she does not have a good relationship with her mother, she is relying on her mother’s financial wealth. In addition, adult daughters’ dependence on their mothers was described in contrast to their complaints about the current childcare policies, which led to their preference for their mothers (*N* = 12). For example, Seung-eun, who has a boy five years old, relied on her mother’s childcare:


*I did not feel that I got much help even if I got support from the government for the cost of daycare education or full-day care. I can ask for my mom at any time I want…but if I use childcare service, I need to follow the application process first, and I do not even know whether the care provider is a trustworthy person or not. So, I prefer my mom. It’s better to pay her not childcare center. I hope the government can give benefit to the household where the grandparents are taking care of grandchildren.*


In contrast, despite the high reliance on their mothers, the adult daughters also prioritized independence for themselves or their families (*N* = 15). Adult daughters described independence in three ways: (1) appropriately compensating the mothers for childcare (*N* = 6); (2) planning future children without relying on their mothers (*N* = 5); (3) feeling free from taking care of their mothers (*N* = 4).

First, adult daughters felt ambivalence when they compensated their mothers for childcare to save money for their family. For example, Joo-young, whose mother is taking care of her daughter all day, is skeptical about compensation for her mother although she depends entirely on her mother’s childcare. She thought it would be more important to make her family independent than to give a small amount of money for her mother:


*All child-care is only from my mom not from other people. She takes care of my daughter about 10 h per day. […] However, it is not much helpful for recovering my mom’s expenses with a small amount of pocket money, I believe it is not necessary to give her. As off-spring’ perspective, my husband and I must maintain our family budget stable through financial activities.*


Similarly, Jeong-a, whose mother is taking care of her two sons, stated she was satisfied with the current childcare expenses, unlike her sisters who were reluctant to pay more for their mothers. She even assumed that the amount would satisfy her mother, and did not want to pay more in order to keep her family financially stable.

Next, adult daughters currently rely on their mothers’ childcare, but the adult daughters’ personal lives are more important for a birth plan. Many adult daughters (*N* = 17) stated that their current mother’s help affected their birth plans. Several adult daughters claimed that if they could not depend on their mothers’ childcare, they would not have more children (*N* = 9). Other adult daughters acknowledged the dependence on their mothers’ help to decide the future birth plan, but they thought their decisions were more important (*N* = 12). For example, Yoon-a, whose mother is taking care of her daughter 12 h per day, stated that her mother’s care for her child would have a major impact when planning for a second baby. However, she hoped to be able to take care of her future child by herself. Some adult daughters thought that they would not have any children even if they currently depend on their mothers’ help because their personal lives are more important (*N* = 6). For example, Tae-ri, living with her mother, relies on her mother’s help for two sons and has no more birth plans because of her life and economic reasons:


*My mom took care of my first child and she told me she would take care of the second child as well, that’s why I had the second baby. […] Even if my mom says she will take care of next baby, I would not plan for the baby. Because raising a child is costly and I feel I might lose my own life, and I do not want it.*


Finally, adult daughters highly depend on their mothers’ help, but they do not desire to take care of their mothers as they view this as a burden (*N* = 6). Some adult daughters transferred responsibility of caring for mothers to their siblings. For example, Ji-min, who lives with her mother during the weekdays, said she could not do anything without her mother, but it also felt like taking care of her mother is a burden:


*My mom is a significant person to me because I can’t do anything without my mom now. On the other hand, there are parts where my mom is a burden for me. She is not ready for her retirement, so I think I should do it for her. Frankly speaking, it is hard to take care of just myself. I don’t want to take care of my mother.*


In sum, adult daughters felt dependence (childcare, emotional, and financial) on their mothers and desired independence (compensation, birth plan, and burden of caring). The “dependence” linked with functional solidarity and the “independence” connected with normative solidarity. Adult daughters who received childcare, emotional, or financial support from their mothers tended to provide support such as money, groceries, serving meals to their mothers. In addition, adult daughters who desired independence from their mothers recognized their family obligation, but hoped to be free from the obligation of caring for their aging mother. Furthermore, when the adult daughters described dependence, they preferred their mothers providing childcare rather than support from the current policy.

### 4.3. Closeness vs. Disagreement

This study found that respondents felt not only closeness to but also disagreement with their mothers. Most respondents (*N* = 19) were (1) living in close proximity at most 30 min driving, (2) almost daily communicating with their mothers, (3) talking about mainly childcare, and (4) felt closeness by frequent interactions with their mothers. Interestingly, only five respondents had close geographic proximity with their parents-in-law, but they reported relatively less contact with their parents-in-law compared to their mothers. For example, even Hye-jin lives within 10 min by car from both her mother’s home and mother-in-law’s, she has not often contacted her mother-in-law. Some adult daughters felt more intimacy with their mothers by moving closer to their mothers’ home (*N* = 8). For example, Seo-jin, who moved to nearby her mother’s home, described her closeness with her mother:


*We meet every day because we are living in the same apartment as the upper and lower level. As I moved to the apartment where my mom lives in, I could do more things with my mother such as cooking, trips, and eating out. I often call and message to mom; even there is nothing special to talk. I am satisfied with communication with my mom. In addition, my daughter’s daycare center is very close from here. It’s free. I am satisfied with it. When my first daughter was young, the government didn’t provide it.*


In contrast, three adult daughters were living together with their mothers, and two families were living together on weekdays and separated on weekends. Although some adult daughters who were living together with their mothers, they reported having low communication frequency and satisfaction.

When communicating about childcare, some adult daughters and their mothers had the same opinions on childcare (*N* = 7). Adult daughters’ values on childcare were indirectly described through the interaction between their mothers and their children. Respondents were satisfied with their mothers providing childcare because they had similar parenting views.

In contrast, other adult daughters in frequent contact with their mothers felt inconvenience due to disagreements regarding parenting styles (*N* = 10). For example, Na-young, shared her experience:


*I see my mother almost every day. I often talk (childcare) on the phone […]. When my mom comes, she stays nearly all day. I was dissatisfied with my mom’s hygiene idea. I used something to wipe the floor, and she washed my child’s hands with it, or when she used my child’s cup again and again, and then I nagged her, she said, “It’s okay. That won’t kill your child,” but I didn’t like it.[…] but she ignored it, and I scolded her.*


In sum, adult daughters felt not only closeness to but also disagreement with their mothers. The “closeness” linked structural, associational, and consensual solidarity, whereas the “disagreement” connected with consensual solidarity. Generally, the adult daughters tended to have a good relationship with their mothers when they had close proximity, frequent contact, and similar values. However, the closeness did not guarantee positive relationships with their mothers. The respondents did not mention much about current childcare policy, unlike the other two themes.

## 5. Discussion

The purpose of this study is to elucidate the experiences of Korean working adult daughters who are supported by childcare from their mothers. Findings provide a further understanding of how working adult daughters perceived the relationship with their mothers who provide childcare.

First, the three themes in this study explained the ambivalence of the adult daughters. All six aspects of the intergenerational solidarity model regarding the relationship between adult daughters and their mothers were found during the data analysis. Adult daughters presented both dependence on their mothers’ assistance (i.e., childcare, emotional, and financial reliance; functional solidarity) and independence (i.e., compensation as related to functional solidarity and the burden of care for mothers as related to normative solidarity).

Furthermore, they felt not only closeness with their mothers (i.e., associational solidarity, structural solidarity, consensual solidarity), but also disagreement with their mothers (i.e., consensual solidarity). However, the three emerged themes were not exactly matched to the six dimensions of the Intergenerational Solidarity Model. For example, “gratitude” and “guilt” did not fit well with any dimensions of the model. Bengtson and colleagues [25] suggested that the intergenerational solidarity-conflict model can cover the concept of ambivalence. Bengtson et al. [25] assumed that a high score on solidarity would suggest an absence of or a low level of conflict. However, the current study did not fit well with this structure; hence, this empirical study suggests that the “intergenerational solidarity-conflict-ambivalence model” would be better to explain intergenerational relationships in adulthood.

Although previous studies also found the adult daughters’ ambivalent feelings toward their mothers in relation to family-provided childcare [9,10], they did not link the cause of ambivalence to intergenerational solidarity. Theoretically, the various dimensions of intergenerational solidarity measuring the behavioral consequences of intergenerational relationships (i.e., contact, resource exchange, the degree of agreement, etc.) are different from the ambivalence measuring the perception of relationship quality [26]. However, since the behavior of the intergenerational relationship and the quality of relationship are highly related to each other [27], the current research extends the previous studies.

Second, the three themes reflected both psychological ambivalence and sociological ambivalence. According to Lüscher and Pillemer [28], intergenerational ambivalence consists of psychological ambivalence and sociological ambivalence. Psychological ambivalence means that individuals experience conflicting emotions, positive and negative, at the same time. Sociological ambivalence reflects the incompatibility of the demands expected in specific social roles. The three emergent themes indicate adult daughters’ individual conflicting emotions toward their mothers as psychological ambivalence. For example, many adult daughters were experiencing both gratitude and guilt while being supported by their mothers. This is a conflicting emotion within the individual; however, the cause of these emotions is relevant to the normative role given to the adult daughters. When parents and children have different norms, parents and children experience psychological ambivalence [28]. Affection and conflict between parents and children can be more intense in a strong familial culture [29].

In Korea, although strong Confucian values remain [9], family functions are rapidly changing. In other words, traditional family norms (i.e., filial piety, the idea that a married daughter was no longer a member of their own family) and modern family norms (i.e., personalization, egalitarianism) coexist [27]. From a traditional point of view, childcare from mothers would be unnecessary for married adult daughters, however they depend on the help of their mothers in the real world. From another traditional perspective, adult daughters should take care of their aging mothers, but rather they currently receive help from their mothers. The adult daughters who have those two traditional family norms may experience a sense of guilt toward their mothers who provide childcare. Meanwhile, adult daughters feel a sense of gratitude to their mothers because they can build their careers and keep their personal lives due to their mothers’ support. Adult daughters may feel a sense of ambivalence because they try to balance the collective desires of family well-being and personal needs (e.g., autonomy) [30]. This supports the view that psychological ambivalence is likely to arise in the presence of social ambivalence [31].

Third, the adult daughters’ ambivalence reflects the lack of alternative options. Most of the participants felt emotionally closer to their mothers after the first childbirth, and they felt even closer when the mothers started providing childcare. This confirms the findings of Kaufman and Uhlenberg [32], that adult daughters felt much closer to their mothers when they frequently visited or called them. However, more frequent visits or calls did not guarantee a good relationship. Adult daughters had a high level of contact with their mothers, however lacked exit options in two respects: geographic proximity and dependence on childcare. First, most of the adult daughters, except for five, lived in close proximity to their mothers (within 30 min driving distance). Adult daughters were able to experience closer contact because they lived nearby; however, the proximity increased interference of mothers into their adult daughters’ parenting. These discrepancies in opinions caused the ambivalence feelings for the adult daughters. Second, adult daughters lacked other alternatives for childcare in addition to their mother. The absence of nearby family members (i.e., siblings, parents-in-law) who could provide childcare forced the adult daughters to rely on their mothers. As adult daughters have been dependent entirely on their mothers for childcare, they expect to negotiate with their mothers on compensation. In this process, the ambivalence of the adult daughters increased because of the idea of securing the independence of their home economics and the guilt of not giving more money to their mothers. Those findings support the fact that reduced alternative options contribute to negative ambivalence in close relationships [33].

Lastly, the ambivalent feeling of adult daughters in Korea consistently existed despite the reform of the childcare policy in 2013. The reformed childcare policy plays a role as a macro factor impacting the relationship quality between working adult daughters and their mothers. Some adult daughters welcomed the policy. Generally, adult daughters who had stable jobs with good childcare benefits, such as long maternity leave, tended to have no complaints about the current childcare policy. In addition, in the case of adult daughters who did not experience benefits when their first child was born but their second child is under the policy, they were satisfied with the current childcare policy. However, most of them pointed out the ineffectiveness of the policy (i.e., absence of trustworthy persons, low quality of childcare centers, too small amount of benefits). This perceived ineffectiveness became an excuse to receive their mother’s childcare support. The main reason why the adult daughters relied on their mothers was that their mothers were more reliable than other providers in childcare centers. They felt gratitude for their mothers’ childcare support, but felt guilt as well. In addition, the adult daughters’ working climate was related to the ineffectiveness of the childcare policy. The adult daughters explained that it is more important to lessen working hours or have flexible working time rather than implementing childcare policy. Due to long working hours and rigid working time, the adult daughters had to ask their mothers to take care of their children for a longer period of time. This increased their feelings of guilt toward their mothers. Furthermore, the current childcare policy did not influence the adult daughters’ birth plan. Rather, the decision to rely on their mothers for childcare was the more practical option for them.

The adult daughters reported the same preference for an informal childcare system rather than formal childcare system when the children are younger, regardless of the change of childcare policy. Within this cultural context, this study along with previous empirical studies found that adult daughters felt both gratitude and guilt toward their mothers who provided childcare [9,10]. This raises the need to examine the effectiveness of current government policies more closely. The government needs to reconsider the fact that adult daughters are being helped by their mothers despite the feelings of guilt. It may reflect that adult daughters distrust the aid from others or public service. In addition, the government should examine the real needs of dual income families when establishing policies. For example, working adult daughters depend on their mothers to fill the childcare deficit caused by the gap between their commuting time to work and their children’s commuting time to the childcare center. This is not a problem that can be solved by subsidy or childcare allowance. Not only improving the quality of public childcare service but also diversification of services to reflect the needs of dual-income families will be required.

### Limitations and Future Directions

Limitations of this study should be considered for future research. First, this study investigated the intergenerational relationship experiences with working adult daughters in depth, however a significant limitation is not examining the perspectives of adult daughters’ mothers. According to the intergenerational stake hypothesis [34], parents may be expected to feel less ambivalence than their children because the parents perceive parent–child relationships more positively than their children. Previous studies have supported this hypothesis that adult daughters who have a higher intergenerational dependency tend to feel a higher sense of ambivalence toward their mothers while mothers tend to feel a higher sense of positive emotions toward their adult daughters [35]. However, it is necessary to conduct further empirical research to see if this finding is replicable in other cultures. The dyad relationship would be a comprehensive study to understand older mother-adult daughter relationship quality.

Second, future research should consider family dynamics, outside of the mother-daughter ties, that could impact adult daughters’ emotions. For example, adult daughters can feel a different kind of ambivalence under dynamics between mothers and their husbands. This study could see an indirect relationship between mothers and their sons-in-law through adult daughters, however, it did not explore the real dynamics between mothers-in-law and sons-in-law. Sons-in-law may feel gratitude toward their mothers-in-law due to the dependency of childcare and can feel discomfort toward them due to too much interruption of mothers-in-law. These dynamics within the close family of adult daughters may influence adult daughters’ relationship quality with their mothers.

Finally, future studies should examine the relationship between adult daughters and their mothers from various socio-economic backgrounds. This study’s sample is largely comprised of working women in their 30s with a high level of education. This highly educated sample might cause selection bias. Eighteen mothers of the adult daughters in this study were from the Baby boomer cohort in Korea (1955–1963), which is considered a powerful middle-class force in their peak income period. Their relationship ties may have different dynamics compared to ties with disadvantaged demographics.

Despite the limitations, this current study contributes to expanding the limited research on the relationship dynamics between Korean working adult daughters and their mothers who provide childcare through applying the intergenerational solidarity model. Furthermore, the current study contributes to further understanding of how the components of the macro level (i.e., Korean current childcare policy reformed in 2013) affect the relationship quality between grandparents and their adult children.

## 6. Conclusions

The current study explored how interpersonal, familial, and contextual processes contribute to Korean working adult daughters’ perception of their current relationship with their mothers, who provide supplementary grandchild care. Findings from interviews revealed that adult daughters had three ambivalent feelings regarding their mothers’ childcare provision. First, the respondents felt both gratitude and guilt toward their mothers. Second, participants expressed feelings of dependence on their mothers but also desired independence. Lastly, adult daughters felt not only closeness to but also disagreement with their mothers. The ambivalent feelings of adult daughters in Korea have consistently existed despite the reform of the childcare policy in 2013. In other words, all the adult daughters were receiving universal childcare subsidies, but it was not enough to meet the needs of working adult daughters. This study extended previous studies on childcare by combining multiple intergenerational solidarities. All six aspects of intergenerational solidarity in the relationship between adult daughters and their mothers were found during the data analysis, but three themes emerged through interconnected dimensions. The findings imply that more empirical studies should be explored at both the macro and micro-level in family studies.

## Figures and Tables

**Table 1 ijerph-19-14226-t001:** Sample Socio-Demographic Characteristics.

	Pseudonym	Age	Occupation	Level of Education	Income (KRW)	Mother’s Age
1	Min-a	39	Police officer	Bachelor’s degree	7,000,000 (=$6113)	65
2	Jeong-a	36	Tax officer	Bachelor’s degree	6,500,000 (=$5676)	60
3	Jin-sol	35	Cashier	Bachelor’s degree	5,500,000 (=$4803)	61
4	So-mang	35	College assistant	ABD	11,750,000 (=$10,262)	58
5	Joo-young	35	Trade officer	Bachelor’s degree	6,000,000 (=$5240)	62
6	Dan-i	35	Lecturer	ABD	3,500,000 (=$3056)	60
7	Hye-na	36	Automobile officer	Associate degree	6,000,000 (=$5240)	64
8	Se-ri	43	Director of kindergarten	ABD	7,000,000 (=$6113)	68
9	Yoon-a	33	Bank clerk	Bachelor’s degree	7,000,000 (=$6113)	58
10	Ga-yun	35	Trade officer	Bachelor’s degree	5,000,000 (=$4366)	61
11	Hong-you	36	Insurance planner	Associate degree	4,250,000 (=$3711)	69
12	Joo-ha	32	Businessman in baby food	Bachelor’s degree	500,000 (=$3930)	57
13	Seung-eun	36	Social worker	Bachelor’s degree	5,500,000 (=$4803)	68
14	Na-young	38	Social worker	Bachelor’s degree	5,000,000 (=$4366)	64
15	Ji-min	42	Nurse	Bachelor’s degree	5,000,000 (=$4366)	66
16	Hye-jin	30	Social worker	Bachelor’s degree	5,300,000 (=$4628)	54
17	Tae-ri	37	Teller	Associate degree	5,500,000 (=$4803)	62
18	Young-mi	37	Health teacher	Bachelor’s degree	6,250,000 (=$5458)	67
19	Ye-seo	35	Managerial clerk	Bachelor’s degree	10,000,000 (=$8733)	75
20	Seo-jin	36	Elementary teacher	Master’s degree	6,500,000 (=$5676)	61
21	Mi-hyang	36	Child psychologist	Master’s degree	5,000,000 (=$4366)	58
22	Na-ra	31	Finance officer	Bachelor’s degree	6,000,000 (=$5240)	57
23	Soo-im	34	Elementary teacher	Bachelor’s degree	5,500,000 (=$4803)	62
24	Seung-hye	35	Elementary teacher	Master’s degree	5,500,000 (=$4803)	61

Note: ABD (All However, Dissertation)

## Data Availability

Not applicable.

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
