# Peer review of "The Role of Maternal Grandmothers’ Childcare Provision for Korean Working Adult Daughters"

_ijerph, 2022, doi:10.3390/ijerph192114226_

Round 1
Reviewer 1 Report
Overall, the manuscript is well written and both rationale and methods are sound. The conclusions further add to the overall knowledge on such a delicate and increasing relevant topic within the Western society. There are, nevertheless, some minor comments regarding the Method and Findings sections, detailed below: Method section Participants subheading - when referring to mean age, the numbers should also be accompanied by the standard deviation - the authors should refer $ as US$ - the need for attribution of pseudonyms should be mentioned as to guarantee the protection of the participants - Table 1: the column Income is not clear. I don't understand what Income means (does it means year / month / week?). Also, with such large numbers (which need to be multiplied by 10,000), it does require some cognitive effort to interpret. Findings section - there are some transcripts missing following the text (e.g., page 5, line 201; page 8, line 330) Gratitude vs Guilt subheading - the transcript between lines 183-189 needs proper formattingAuthor Response
First of all, we sincerely appreciate the time and efforts you have provided. Your thoughtful suggestions have helped us substantially improve on our original submission. Please find our detailed response to your comments in the attached file.

Reviewer 2 Report
Table 1 and related text in the method section might be better placed in finding the section
It is highly recommended to provide charts or tables in the finding section in order to summarize this section.
Author Response
First of all, we sincerely appreciate the time and efforts you have provided. Your thoughtful suggestions have helped us substantially improve on our original submission. Please find our detailed response to your comments in the attached file.

Reviewer 3 Report
Using 24 semi-structured interviews and the Intergenerational Solidarity Model, this study examines the experiences of adult daughters with their mothers' informal child care assistance. The article is extremely well-written and engaging. The research is clearly rooted in intergenerational solidarity's theoretical perspectives. By presenting novel evidence and arguments, it fills a distinct knowledge gap in our understanding of the intersections between child care assistance and adult daughters' reflections. Despite the fact that I do not have many comments because the manuscript has already done an excellent job, a few revisions would strengthen the paper.
1) Abstract: The following themes are presented in the abstract: “gratitude vs. guilt, reliance vs. independence, and intimacy vs. conflict.” Later in the manuscript, a variant version appears: “(1) gratitude vs. guilt, (2) dependence vs. independence, and (3) closeness vs. disagreement.” It would be preferable if there were consistency with the themes throughout the entire paper.
2) Abstract: The abstract ends with the following: “The results indicated that adult daughters were found to have mixed emotions about their mothers' childcare.” This was anticipated and even inevitable. The authors articulated the findings very well in the discussion section; consequently, they could have concluded the abstract with a more comprehensive and compelling argument.
3) Introduction: “To date, research on grandparent childcare has focused on” Could you specify the field of study, such as psychological, sociological, or socio-psychological?
4) General suggestion: The manuscript occasionally criticized previous scholarship with a different theoretical orientation and empirical evidence, which does not make these studies inferior. I respectfully recommend the authors not to "make enemies" by directly criticizing prior research, but rather to present their research as original and unique while acknowledging what previous research did. For example:
“they did not explore adult children’s perspective and supplementary caregiving.”
“the studies were not able to explain the unique dynamics between working-women and their mothers who pro- vided childcare due to lack of specifying the participants (i.e., working adult daughter and daughter-in-law; working adult daughter and non-working adult daughter).”
“However, those two studies lacked con- sideration of the reformed childcare policy in Korea, which may contribute to the relation- ship quality between grandparents and their adult children.”
5) Introduction: It appears that the Universal Childcare Subsidy policy is central to this investigation. Could you explain this policy in a footnote for readers who are less familiar with it and considering that it merits further elucidation?
6) Theoretical framework: “First, the concept of linked lives within the life course perspective will be applied to understand the mother- daughter relationship.” Whose concept is “linked lives”? Although it is argued that this concept will be utilized, we do not encounter this concept again.
7) Theoretical framework: Even if the study employs "Exchange Theory" arguments, it is not mentioned, for example:
“Korean aging mothers also view provid- ing childcare as an investment in their daughters [12].”
“From another traditional perspective, adult daughters should take care of their aging mothers, but rather they currently receive help from their mothers. The”
Dowd JJ (1975) Aging as exchange: a preface to theory. Journal of Gerontology 30, 584–594.
Johnson, C. L. (1983). A cultural analysis of the grandmother. Research on Aging, 5(4), 547–567. https://doi.org/10.1177/0164027583005004007.
8) Theoretical framework: “Furthermore, scholars have found that grandparents who provided childcare expe- rienced both positive emotions [8] and negative emotions [22] at the same time.” Given that there are two studies with "different findings," this argument is ontologically flawed. Either the authors could note that the scholarly research yields mixed results or they could cite a single or multiple studies demonstrating both positive and negative emotions.
9) Max-QDA package -> Max-QDA program.
10) Findings: The first quotation does not adhere to the structure (font, font size, and indentation) of the subsequent quotations.
“I appreciate my mother’s sacrifice. If there was no sacrifice from my mom, I could not work. It means that I might not have any financial ability. Therefore, her sacrifice is also helpful for my financial status. Of course, the government supports my baby for $150 per month. To my knowledge, it will continue only 2 years. I don’t think it’s helpful. Ac- tually, it is my social right due to paying tax. Rather than the help, my mother supports me a lot. It is a great thing that there was no cut off time in my career and it’s all due to my mom.”
11) Findings: “which led to their preference for their mothers(N=12)” Space is needed after “mothers.”
12) Findings: Right before the Discussion, various types of solidarities, namely structural, associational, and consensual, are mentioned. Please explain them in the theoretical framework section.
Author Response

(The authors gave the same response as above.)
